# Methodological considerations in the assessment of direct and indirect costs of back pain: A systematic scoping review

Dawit T. Zemedikun[1]*, Jesse Kigozi[1], Gwenllian Wynne-Jones[2], Alessandra Guariglia[3], Tracy Roberts[1]

1 Health Economics Unit, Institute of Applied Health Research, University of Birmingham, Birmingham, England, United Kingdom, 2 Primary Care Centre Versus Arthritis, Research Institute for Primary Care & Health Sciences, Keele University, Staffordshire, Keele, England, United Kingdom, 3 Department of Economics, University of Birmingham, Birmingham, England, United Kingdom

* D.T.Zemedikun@bham.ac.uk

**Data Availability Statement:** All relevant data are within the paper and its Supporting information files.

## Abstract

### Background

Back pain is a common and costly health problem worldwide. There is yet a lack of consistent methodologies to estimate the economic burden of back pain to society.

### Objective

To systematically evaluate the methodologies used in the published cost of illness (COI) literature for estimating the direct and indirect costs attributed to back pain, and to present a summary of the estimated cost burden.

### Methods

Six electronic databases were searched to identify COI studies of back pain published in English up to February 2021. A total of 1,588 abstracts were screened, and 55 full-text studies were subsequently reviewed. After applying the inclusion criteria, 45 studies pertaining to the direct and indirect costs of back pain were analysed.

### Results

The studies reported data on 15 industrialised countries. The national cost estimates of back pain in 2015 USD ranged from $259 million ($29.1 per capita) in Sweden to $71.6 billion ($868.4 per capita) in Germany. There was high heterogeneity among the studies in terms of the methodologies used for analysis and the resulting costs reported. Most of the studies assessed costs from a societal perspective (n = 29). The magnitude and accuracy of the reported costs were influenced by the case definition of back pain, the source of data used, the cost components included and the analysis method. Among the studies that provided both direct and indirect cost estimates (n = 15), indirect costs resulting from lost or reduced work productivity far outweighed the direct costs.

**Funding:** The authors received no specific funding for this work.

**Competing interests:** The authors have declared that no competing interests exist.

## Conclusion

Back pain imposes substantial economic burden on society. This review demonstrated that existing published COI studies of back pain used heterogeneous approaches reflecting a lack of consensus on methodology. A standardised methodological approach is required to increase credibility of the findings of COI studies and improve comparison of estimates across studies.

## Introduction

Cost of illness (COI) studies aim to identify and measure the economic impact of an illness including the direct and indirect costs [1, 2]. These studies are descriptive, and they quantify the costs without comparing alternative uses of healthcare resources. COI studies can, however, serve as a basis for further economic evaluations and they are particularly useful for chronic diseases that impact heavily on health expenditures [3, 4].

Back pain is a common health problem and is the leading cause of years lived with disability in most countries and age groups [5–7]. Although back pain has low impact in terms of mortality, it imposes great medical and non-medical related costs on patients, employers, and health care providers [1, 8, 9]. As society is aging worldwide, these costs are likely to rise putting further pressure on health care services.

Despite the substantial costs reported in COI studies of back pain, there is little guidance in the literature to support the choice of methodologies in those studies. Dagenais et al. [1] conducted a systematic review of COI studies in LBP focussing on the magnitude of the economic burden rather than the methodologies. Their review from 2007 was restricted to one biomedical database (Medline) and examined only studies published in the previous 10 years. In the intervening years, a number of studies have emerged across many countries. There is a clear lack of consensus about the appropriate methodologies to use to estimate the economic impact of back pain. The objective of this review was to systematically gather and characterise the body of literature on the direct and indirect costs of back pain in order to evaluate the methodological approaches used by researchers in developing COI studies of back pain. We also present the resulting national estimates of direct and indirect costs of back pain from the reviewed studies.

## Methods

We conducted a systematic scoping review guided by the framework introduced by Arksey and O'Malley [10], and following the Preferred Reporting Items for Systematic Reviews and Meta-Analyses (PRISMA) [11] guidelines for reporting.

### Search strategy

Six electronic databases (MEDLINE, Embase, CINAHL Plus, Web of Science, EconLit, and Centre for Reviews and Dissemination (CRD)) were searched for studies published in English from inception to February 2021. We focussed the search on OECD countries where access and structure of the healthcare systems are more comparable amongst these high-income countries. The literature searches were conducted using a combination of keyword searching and medical subject headings (MeSH). The searches were made robust by making use of wild-cards, phrase searching and truncation of the search terms as appropriate (S1 File). Only full

text articles were considered ensuring that studies with sufficient methodological detail were assessed.

**Inclusion and exclusion criteria.** Studies had to meet the following inclusion criteria:

1. Conducted in the UK and other high income countries (OECD members [12]).

2. Concerned with economic burden of back pain or low back pain.

3. Cost provided as monetary estimate of direct or indirect costs.

4. Studies investigating adult patients.

5. Reports written in English.

The exclusion criteria used were as follows:

1. Musculoskeletal conditions other than back pain.

2. Economic evaluations of interventions.

3. Review articles.

4. Abstracts or conference proceedings.

**Selection of studies for review.** Eligibility of the identified studies was assessed using a two-stage categorisation process (Table 1) that have been described in detail elsewhere [13–15]. The categorisation process was designed to be as inclusive as possible so that no study fell outside of the predefined groups. One reviewer (DZ) screened studies initially by title and then categorised them into six groups (A-F). This was followed by full text reading of potentially

**Table 1. Categorisation process for selection of studies for review.**

| |
|---|
| **Stage I—Initial categorisation of studies**: |
| A. The study reports primary or secondary research on the economic burden of back pain and provides substantial cost data. |
| B. The study discusses the cost of back pain and provides estimates of some aspects of COI or components of direct or indirect costs. |
| C. The study provides useful information on assessing the economic burden of back pain but does not entirely fall into either A or B. (e.g. methodological studies on COI without reporting direct or indirect costs estimates). |
| D. The study discusses general aspects of the economic impact of back pain but provides little or no data on direct or indirect costs (e.g. economic evaluations). |
| E. Full text of the study is not available (abstracts, conference proceedings). |
| F. The study does not have any relevance to the economic burden of back pain. |
| *Studies in category (A), (B) or (C) will deemed relevant for the systematic review while those in category (D), (E) or (F) were excluded at this stage.* |
| **Stage II—Further categorisation of studies**: |
| 1. Cost of illness (COI) analysis studies (direct or indirect cost) |
| 2. Other cost studies |
| 3. Description of methods used in assessing cost of back pain |
| 4. Private out of pocket expenditure |
| 5. Economic evaluations |
| 6. Review articles without new data |
| 7. Not relevant for economic burden of back pain. |
| *Studies classified as A(1), A(2), B(1) and B(2) were determined to be suitable for data extraction. Studies coded as C (3) and ABC(6) were retained for background literature and discussion purposes. All other studies not classified into one of the above categories were excluded.* |

relevant studies and further classified them into eight groups (1–8). A second reviewer (JK) retrieved and reviewed a random sample of 25% of the studies at each stage of the selection process to assess agreement. Any discrepancies were resolved by discussion until a consensus was reached by all five authors. We did not exclude papers on quality grounds as the purpose of the review was to identify the range of methods that have been used to estimate the cost of back pain and to identify the entire range that has been apportioned to back pain, in order to inform a planned primary study.

For each study, a range of data including study characteristics, methodology used, and the results reported were extracted using an electronic template. The information was tabulated, and the methodology and findings of individual studies were compared narratively. For consistency and standardisation across studies, all costs were converted to 2015 US dollars using country-specific gross domestic product inflator index and purchasing power parity (PPP) conversion [16].

## Results

The search identified a total of 8,009 potential citations. After removing duplicates and title screening of the citations, 1,588 studies reached the initial categorisation stage. Following the initial categorisation by title and abstract, 55 studies were included in the second stage of the review process. These articles were read in full and further classified to determine their suitability for inclusion resulting in a total of 45 studies coded A(1), A(2), B(1), and B(2) that met the criteria for the review (Fig 1).

The 45 studies included 17 studies (39%) from the United States [17–33], five (11%) from Sweden [34–38], four (9%) from The Netherlands [39–42], and three (7%) each from the UK [9, 43, 44], Germany [45–47], and Japan [48–50] (Table 2). The studies were published from 1995 to 2020, and the data collection spanned from 1987 to 2017. The age of data at the time of publication in the reviewed studies ranged from one year [30] to 11 years [18, 28]. There was also high heterogeneity among the studies in terms of the methodologies used for analysis and the resulting cost estimates reported.

The main methodological characteristics of the included studies are summarised below. A general description of the introductory concepts and approaches used in COI studies is first given (Table 3).

### Back pain diagnostic criteria

International Classification of Diseases codes (ICD-9 or ICD-10 codes) or a variant of these were typically used to define back pain by over half of the studies, while self-reported back pain assessment was reported in 15 studies [18–20, 26, 28, 35, 39, 45, 47, 49, 51–55]. The diagnostic criteria used was not explicitly specified in some of the studies [19, 30, 50, 56, 57]. However, studies with diagnostic codes were often used to produce national cost estimates compared to studies with self-reported or non-specific back pain definition types that mainly provided either average costs per patient or indirect cost estimates.

### Source of data

A diverse range of data sources were used including surveys, national databases, compensation claims and cost diaries. Multiple sources of data were typically used in many of the studies, and the reported direct or indirect costs were the result of combining and summing an array of data sources. Large-scale surveys, and claims databases dominated the source of utilisation data with the latter being used mainly in insurer perspective studies. The use of electronic health records (EHRs) and registry data was limited [34, 37, 38, 41, 43]. In the US, the Medical

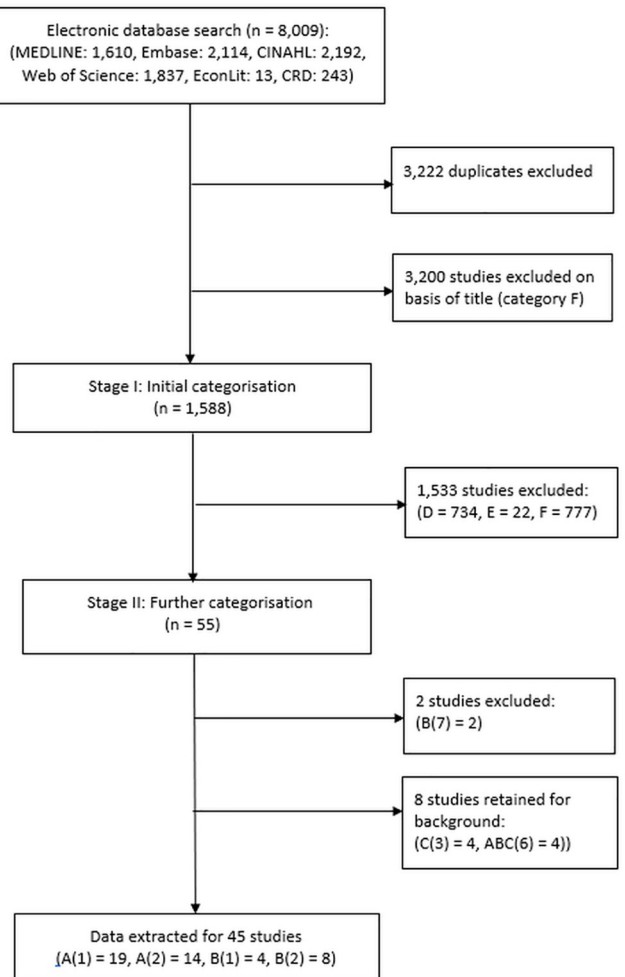

**Fig 1. PRISMA diagram of literature search and study selection.**

Expenditure Panel Survey (MEPS) database was the single most common source of data for analysing COI for back pain [22, 24, 28, 29].

## Perspective of the analysis

The societal perspective which is preferred since it is the most comprehensive perspective was adopted by the majority of the studies (n = 29) while only two studies followed a healthcare perspective [43, 51]. Taking the viewpoint of insurance-based health care, the insurer perspective was the second most popular [17, 20, 21, 23, 25, 27, 31, 32, 46, 50, 56] perspective. It was apparent that most of the insurer perspective studies (9 out of 12) were US studies [17, 20, 21, 23, 25, 27, 31–33] conducted using claims databases. Cost analysis from industry or employer perspective is limited in its scope and was less common [18, 19].

## Cost components

No significant methodological differences were observed between studies that assessed both direct, and indirect costs (n = 23) and those that reported on direct costs only [17, 21–23, 25, 27, 29, 31, 43, 46, 48, 51, 56] or indirect costs only [18, 20, 26, 28, 30, 40, 44]. This review

**Table 2. Summary of the main characteristics of the included studies.**

| Lead author, year | Country/ perspective | Population | Study Design | Main data source, year of data | Back pain case definition | Direct cost estimation | Indirect cost approach |
|---|---|---|---|---|---|---|---|
| Walker, 2003 | Australia/Societal | National | R, PB | Australian adult LBP prevalence survey, 2001 | Diagnostic code | Top-down | Human capital & friction cost |
| van Zundert, 2005 | Belgium/Societal | National | R, PB | IDEWE (workers welfare body), 1999 | Other/non-specific | Top-down | n/s |
| Coyte, 1998 | Canada/Societal | National | R, PB | Ontario Health Survey data, 1990–94 | Diagnostic code | Top-down | Human capital |
| Hemmila, 2002 | Finland/Societal | Regional | R, PB | Social Insurance Institution files, and patient records, 1994 | Self-reported | Bottom-up | Human capital |
| Depont, 2010 | France/ Healthcare provider | National | R, PB | Surveys/questionnaires, 2001 | Self-reported | Bottom-up | n/a |
| Muller-Schwefe, 2011 | Germany/Insurer | Health insurer ≈ 5.2 million members | R, IB | German statutory health insurance fund (DAK) claims data, 2006 | Diagnostic code | Bottom-up | n/a |
| Wenig, 2009 | Germany/Societal | National | R, PB | Postal survey by German Back Pain Research Network (GBPRN), 2003–06 | Self-reported | Bottom-up | Human capital |
| Becker, 2010 | Germany/Societal | Regional | P, PB | Cross sectional sample from an RCT, 2004 | Self-reported | Bottom-up | Human capital |
| Watson, 1998 | Isle of Jersey/ Societal | National | P, PB | Social Security database, 1994 | Other/non-specific | n/a | n/s |
| Montgomery, 2017 | Japan/Societal | National | R, PB | Japan National Health & Wellbeing Survey (NHWS), 2011 | Self-reported & diagnosed | Bottom-up | Human capital |
| Itoh, 2013 | Japan/Societal | National | R, PB | Survey of Medical Care Activities in Public Health Insurance, 2011 | Diagnostic code | Bottom-up | n/a |
| Shinohara, 1998 | Japan/Insurer | National | R, PB | Labour Standards Inspection Office claims database, 1991–95 | Other/non-specific | Bottom-up | n/s |
| Kim, 2005 | Korea/Insurer | National | R, IB | Korea Labor Welfare Corporation, 1997 | Other/non-specific | n/s | n/a |
| Olafsson, 2018 | Sweden/Societal | Regional | R, PB | Administrative database VEGA, 2008–11 | Diagnostic code | Bottom-up | Human capital |
| Ekman, 2005 (b) | Sweden/Societal | Regional | R, PB | Surveys/questionnaires, 2002 | Self-reported | Bottom-up | Human capital |
| Ekman, 2005 (a) | Sweden/Societal | National | R, PB | Survey and registry data, 2001 | Diagnostic code | Top-down | human capital |
| Hansson, 2005 | Sweden/Societal | Regional | P, PB | Prospectively entered diaries and questionnaires, 1994–95 | Diagnostic code | Bottom-up | Human capital |
| Jonsson, 2000 | Sweden/Societal | National | R, PB, IB | National Board of Health and Welfare's register, 1994 | Diagnostic code | Top-down | Human capital |
| Wieser, 2011 | Switzerland/ Societal | Regional | R, PB | Large population-based survey, 2005 | Self-reported | Bottom-up | Human capital & friction cost |
| Lambeek, 2011 | Netherlands/ Societal | National | R, PB | National registries and authorities, 2007 | Diagnostic code | Top-down | Human capital |
| Boonen, 2005 | Netherlands /Societal | National | P, IB | Cost diaries from three cohorts, 2002 | Self-reported | Bottom-up | Friction cost |
| van Tulder, 1995 | Netherlands/ Societal | National | R, PB | Survey and registry data, 1991 | Diagnostic code | Top-down | Human capital |
| Hutubessy, 1999 | Netherlands/ Societal | National | R, PB | Social Insurance Council data, 1991 | Diagnostic code | n/a | Human capital & friction cost |
| Alonso-Garcia, 2000 | Spain/Societal | National | R, PB | National Health Survey of 2017 (NHS 2017), 2017 | Self-reported | Bottom-up | Human capital |
| Yumusakhuylu, 2018 | Turkey/Societal | National | R, n/s | Surveys/questionnaires, 2011 | Other/non-specific | Bottom-up | Human capital |
| Icatasiotlu, 2015 | Turkey/Societal | National | R, PB | Surveys/questionnaires, 2013 | Self-reported | Bottom-up | Human capital |
| Hong, 2012 | UK/Health-care provider | National | CC, PB | UK General Practice Research Database (GPRD), 2007–09 | Diagnostic code | Bottom-up | n/a |

*(Continued)*

**Table 2.** (Continued)

| Lead author, year | Country/ perspective | Population | Study Design | Main data source, year of data | Back pain case definition | Direct cost estimation | Indirect cost approach |
|---|---|---|---|---|---|---|---|
| Maniadakis, 2000 | UK/Societal | National | R, PB | Office of Population Censuses and Surveys (OPCS), 1997 | Diagnostic code | Top-down | Human capital & Friction cost |
| Kim, 2019 | USA/Insurer | Health insurer ≈ 75 million members | R, PB | MarketScan Commercial Claims Database, 2007–16 | Diagnostic code | Bottom-up | n/a |
| Smith, 2013 | USA/Societal | National | R, PB | Medical Expenditure Panel Survey (MEPS), 2000–07 | Diagnostic code | Bottom-up | n/a |
| Martin, 2008 | USA/Societal | National | CC, PB | Medical Expenditure Panel Survey (MEPS), 2005 | Diagnostic code | Bottom-up | n/a |
| Mehra, 2012 | USA/Insurer | Large regional health insurer | CC, PB | PharMetrics IMS LifeLink claims database, 2006–08 | Diagnostic code | Bottom-up | n/a |
| Gore, 2012 | USA/Insurer | Health insurer ≈ 62 million members | CC, IB | LifeLink Health Plan Claims Database, 2008 | Diagnostic code | Bottom-up | n/a |
| Ricci, 2006 | USA/Societal | National | R, PB | Caremark American Productivity Audit (telephone survey), 2003–04 | Self-reported | n/a | Human capital |
| Stewart, 2003 | USA/Societal | National | R, PB | American Productivity Audit (telephone survey), 2001–02 | Other/non-specific | n/a | Human capital |
| Lind, 2005 | USA/Insurer | Two Washington State companies | R, PB | Health insurance claims data from insurance companies, 2002 | Diagnostic code | Bottom-up | n/a |
| Mapel, 2004 | USA/Insurer | Health insurer with 240,000 members | CC, PB | Lovelace Health Plan (LHP) administrative databases, 2000–01 | Diagnostic code | Bottom-up | n/a |
| Vogt, 2005 | USA/Insurer | Health insurer with 255,958 members | R, PB | UPMC Health Plan claims database, 2001 | Diagnostic code | Bottom-up | n/a |
| Ritzwoller, 2006 | USA/Insurer | Health insurer with > 410,000 members | R, IB | Keiser Permanente Colorado (KPCO) claims database, 1996–2001 | Diagnostic code | Bottom-up | n/a |
| Luo, 2004 | USA/Societal | National | CC, PB | Medical Expenditure Panel Survey (MEPS), 1998 | Diagnostic code | Bottom-up | n/a |
| Rizzo, 1998 | USA/Societal | National | R, PB | National Medical Care Expenditure Survey (NMES), 1987 | self-reported | n/a | Human capital |
| Hashemi, 1998 | USA/Insurer | Insurer with 10% of WC market | R, IB | Claims data from a large insurer, 1996 | Self-reported | n/a | n/s |
| Guo, 1999 | USA/Employer | US industries | R, PB | National Health Interview Survey (NHIS), 1988 | Self-reported | n/a | n/s |
| Williams, 1998 | USA/Insurer | Regional WC insurer | R, n/s | Detailed Claim Information (DCI) database, 1988–92 | Diagnostic code | Bottom-up | n/s |
| Gustafson, 1995 | USA/Employer | Four participating hospitals | P, n/s | Employer records, 1991–92 | Self-reported | Bottom-up | n/s |

n/a = not applicable, n/s = non-specific, P = prospective, R = retrospective, PB = prevalence based, IB = incidence based, CC = matched case-control

LBP = low back pain, RCT = randomised controlled trial, WC = workers compensation, UPMC = University of Pittsburgh Medical Centre

examined the cost components reported with the aim of identifying the most important cost drivers. For direct costs, studies reporting all three major components of inpatient, outpatient, and pharmaceutical costs were summarised for comparison (Fig 2). Outpatient cost was the most important cost driver in the majority of the studies, followed by inpatient cost.

For indirect costs, this review compared studies reporting at least two of the three major cost components of absenteeism, presenteeism and early retirement (Fig 3). Absenteeism which was assessed in all the studies compared was the most relevant cost driver in most studies. Although presenteeism was only assessed in five of the studies, it was found to be the most significant cost component in three of them representing 44%, 70% and 85% of the total indirect costs.

**Table 3. Main concepts and approaches used in COI studies.**

| Type of approaches/Concepts | Description |
|---|---|
| **Cost categories**: *Direct and indirect costs* | *Direct medical cost*: Costs directly related to the disease. Consultations, hospitalisation, medication, diagnostic tests, and accident and emergency services. |
| | *Indirect cost*: Costs due to lost or reduced productivity caused by the disease. Work absence resulting in lost productivity (termed 'absenteeism'), and decreased productivity for those who continue to work (termed 'presenteeism'). |
| **Epidemiological approaches**: *Prevalence-based Vs. incidence based* | *Prevalence-based*: Evaluates costs for all existing cases in a given period. |
| | *Incidence-based*: Evaluates costs by assessing the number of new cases in a given period. |
| **Cost perspectives**: *Societal, health system, industry, individual perspective* | The perspective of the analysis indicates who bears the costs, which in turn determines which costs are to be included in the analysis. |
| **Resource estimation**: *Top-down Vs. bottom-up approaches* | *Top-down*: Measures the proportion of cost attributed to a disease from aggregate figures. Analysis directed from total to lower levels. |
| | *Bottom-up*: Based on actual consumption of resources by referring to records of patients. Analysis directed from individual levels to the total. |
| **Indirect cost estimation**: *Human capital Vs. friction cost methods* | *Human capital (HC)*: Productivity losses are approximated by the value of the individual's earnings assuming that the person would have continued to work in full health. |
| | *Friction cost (FC)*: Uses what is known as the friction period which is the time until another individual from the unemployment pool replaces the worker who is absent due to sickness. The value of productivity losses is then estimated on the basis of the individual's earnings over the friction period. |

## Estimating resource utilisation

The bottom-up approach was a commonly adopted method (n = 29) to calculate the direct costs of back pain compared to the top-down approach [9, 34, 37, 41, 42, 57–59]. The top-down approach was preferred in studies where cost data were widely available from national health statistics. Cost estimations were more detailed in bottom-up studies since individual-level data were aggregated to get the population-level estimates. However, the application of an incremental cost method using a matched-control or econometric methods was limited [17, 22–25, 43].

## Indirect cost estimation

The human capital (HC) approach was typically applied [26, 28, 30, 34–38, 41, 42, 45, 47, 49, 52, 53, 55, 58, 60] in studies that estimated the indirect costs of back pain. The use of both HC and friction cost (FC) approaches was limited [9, 40, 54, 59]. In studies that applied both methods, the HC approach resulted in significantly higher estimates of the indirect costs of back pain. Hutubessy et al. [40] reported the indirect costs of back pain in The Netherlands in 1991 to be $1.5 billion using the FC method; but this increased by three-fold to $4.6 billion under the HC approach.

The annual national cost estimates of back pain from 26 studies that reported national estimates are summarised in Table 4. The total cost estimates in 2015 USD ranged from $259 million ($29.1 per capita) in Sweden to $71.6 billion ($868.4 per capita) in Germany. Direct comparison of costs between the studies is not feasible due to significant differences in the methodologies adopted. In studies that provided both direct and indirect cost estimates

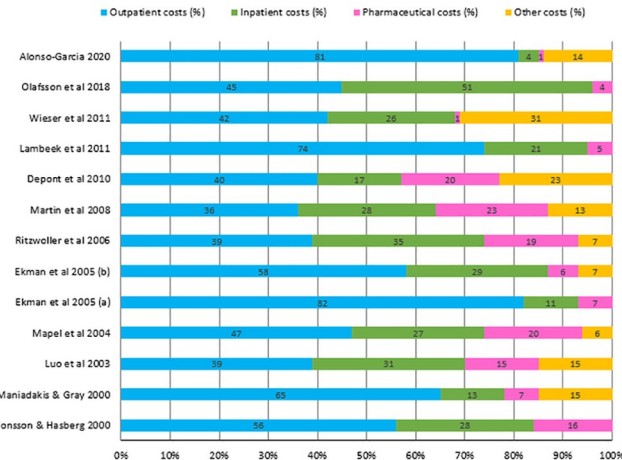

**Fig 2. Allocation of direct costs in COI studies of back pain.** Legend: The figure illustrates the allocation of direct costs in studies that reported on all three major costs components (inpatient, outpatient, and pharmaceutical costs) of direct costs.

(n = 15), indirect costs generally far outweighed the direct costs (S2 File). Measures of precision or dispersion such as confidence intervals or standard deviations around cost estimates were rarely reported in the included studies, and those that reported were largely limited to studies that applied econometric methods for cost estimation [17, 23, 24, 43]. In addition, the measures given were generally for the sample estimates (average resource use or cost) rather than for the extrapolated national cost estimates. This review also uncovered that sensitivity analysis is often not conducted in COI studies with only few studies [9, 24, 39, 43, 47, 54, 58, 59] performing any sensitivity analysis. Moreover, the use of an alternative cost estimation method was considered only in a minority of the studies [9, 24, 40, 54, 59] but resulted in considerably different estimates, particularly for indirect costs. Irrespective of the analysis method used, the reported results reveal the significant economic burden of back pain on healthcare systems and society as a whole.

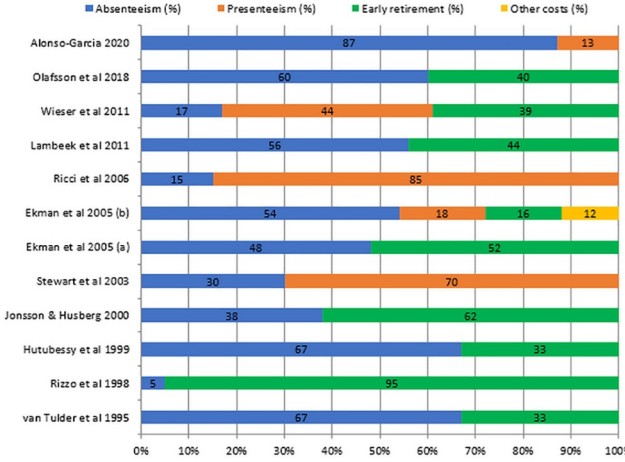

**Fig 3. Allocation of indirect costs in COI studies of back pain.** Legend: The figure illustrates the allocation of indirect costs in studies that reported on at least two of the three major costs components (absenteeism, presenteeism, and early retirement costs) of indirect costs.

**Table 4. National estimates of direct, indirect, and total costs of back pain.**

| Ref. | Country | Population (million) | Direct costs | | | Indirect costs | | | Total costs | |
|---|---|---|---|---|---|---|---|---|---|---|
| | | | National (million $) | % | Per capita | National (million $) | % | Per capita | National (million $) | Per capita |
| [34] | Sweden | 8.9 | 42 | 16 | 4.7 | 217 | 84 | 24.4 | 259 | 29.1 |
| [36] | Sweden | 8.9 | 61 | 15 | 6.9 | 346 | 85 | 38.9 | 407 | 45.7 |
| [38] | Sweden | 9.4 | 261 | 33 | 27.7 | 527 | 67 | 56.0 | 788 | 83.7 |
| [57] | Belgium | 10.2 | 302 | 16 | 29.5 | 1,603 | 84 | 156.7 | 1,905 | 186.2 |
| [41] | Netherlands | 16.4 | 622 | 13 | 38.0 | 4,014 | 87 | 245.1 | 4,636 | 283.0 |
| [55] | Spain | 46.5 | 3,380 | 25 | 72.6 | 9,878 | 75 | 212.3 | 13,257 | 284.9 |
| [48] | Japan | 127.8 | 26,699 | 69 | 208.9 | 11,866 | 31 | 92.8 | 38,565 | 301.8 |
| [58] | Canada | 29.1 | 832 | 8.3 | 28.6 | 9,209 | 92 | 316.4 | 10,041 | 344.9 |
| [9] | UK | 58.5 | 3,363 | 13 | 57.5 | 22,015 | 87 | 86.7 | 25,378 | 433.9 |
| | | 58.5 | 3,363 | 25 | 57.5 | *10,358 | 75 | 177.1 | 13,721 | 234.6 |
| [37] | Sweden | 8.8 | 130 | 3.3 | 14.8 | 3,799 | 97 | 432.7 | 3,929 | 447.5 |
| [59] | Australia | 19.4 | 1,058 | 11 | 54.5 | 8,400 | 89 | 432.8 | 9,458 | 487.3 |
| | | 19.4 | 1,058 | 17 | 54.5 | *5,220 | 83 | 268.9 | 6,278 | 323.4 |
| [42] | Netherlands | 15.1 | 586 | 7.4 | 38.9 | 7,319 | 93 | 485.7 | 7,905 | 524.6 |
| [39] | Netherlands | 16.2 | 6,101 | 66 | 377.8 | 3,206 | 34 | 198.5 | 9,307 | 576.3 |
| [54] | Switzerland | 7.4 | 2,109 | 39 | 283.5 | 3,326 | 61 | 447.0 | 5,435 | 730.5 |
| | | 7.4 | 2,109 | 54 | 283.5 | *1,785 | 46 | 239.9 | 3,894 | 523.4 |
| [47] | Germany | 82.5 | 33,176 | 46 | 402.3 | 38,438 | 54 | 466.1 | 71,614 | 868.4 |
| [48] | Japan | 127.8 | 791 | na | 6.2 | na | na | na | na | na |
| [56] | Korea | 46.0 | 564 | na | 12.3 | na | na | na | na | na |
| [43] | UK | 62.3 | 4,457 | na | 71.6 | na | na | na | na | na |
| [24] | USA | 295.5 | 39,000 | na | 132.5 | na | na | na | na | na |
| | | 295.5 | ~102,000 | na | 346.9 | na | na | na | na | na |
| [22] | USA | 275.9 | 126,258 | na | 457.6 | na | na | na | na | na |
| [26] | USA | 292.8 | na | na | na | 9,115 | na | 31.1 | na | na |
| [44] | Jersey | 0.1 | na | na | na | 3 | na | 36.6 | na | na |
| [18] | USA | 266.3 | na | na | na | 20,287 | na | 76.2 | na | na |
| [28] | USA | 287.6 | na | na | na | 25,559 | na | 88.9 | na | na |
| [28] | USA | 269.4 | na | na | na | 40,318 | na | 149.7 | na | na |
| [40] | Netherlands | 15.1 | na | na | na | 7,339 | na | 487.0 | na | na |
| | | 15.1 | na | na | na | *2,387 | na | 158.0 | na | na |

* Estimated with alternative friction cost (fc) approach for the study above

~ Estimated with alternative incremental cost method for the study above

All costs are presented in 2015 USD.

## Discussion

This review examined 45 COI studies with the aim of assessing the methodologies used to estimate direct and indirect costs associated with back pain. The findings of the review indicated that there is little consensus in the methodologies used to derive cost estimates for back pain, and the reported costs were substantial and wide-ranging.

## Summary of findings

Findings from the included studies confirm that back pain is a costly problem worldwide. The national cost estimates ranged from $259 million to $71.6 billion. It was clear from the studies

reviewed that indirect costs were the main cost drivers for back pain and consequently, major cost savings could be obtained from interventions that bring about early return to work and reduce productivity losses. Outpatient costs were the main cost drivers of direct costs, while absenteeism represented the largest share of indirect costs. However, in the case of indirect costs, our review indicated that presenteeism is often underexplored but represents a significant proportion of the indirect costs of an illness. The significance of presenteeism for the value of lost production were also highlighted in previous literature [54, 61, 62]. A sound methodological framework for the assessment of presenteeism poses a challenge, but the potential impact of presenteeism on costs needs to be included in order to improve the reliability of results [61, 62].

Several factors were likely to have influenced the magnitude and accuracy of the estimates reported. Comparing and generalising these quantitative results is problematic because significantly different approaches had been adopted to estimate the economic burden of back pain. The validity of each method would be related to the available data and the proposed use of the findings. Hence, this review did not find any particular features that should be absolutely avoided to generate valid data. Almost all studies that assessed direct costs reported costs relating to inpatient, outpatient, and pharmaceutical costs; however, few studies included costs relating to emergency department, occupational therapy, or allied health care. Similarly, only few studies included all three major components of indirect costs potentially indicating lack of data source or standardised instruments for the assessment of some indirect cost components. Hence, a wide range of estimates were found across the included studies both nationally and internationally.

## Key methodological challenges

The discrepancies in reported costs and methodologies did not appear to be attributed to the cost perspective taken since most studies adopted the societal perspective. The main sources of variations in the methodologies used in COI studies of back pain were the way in which back pain was defined, the sources of data used, the cost components included, and the approaches used to estimate both direct and indirect costs (e.g. Top-down vs. bottom-up or human capital vs. friction cost methods). There were also considerable discrepancies between the year of data used and the year of publication which should be carefully considered in order to avoid drawing conclusions from outdated data. Quantitative estimates reported should therefore be interpreted with caution taking into account any changes that may have taken place in the time period between pricing and publication.

Diagnosis of back pain should ideally be on clearly defined criteria so that studies might be comparable and replicated as necessary. Consensus on criteria and assessment of the reported cases may be a practical solution for addressing the discrepancies in case definitions [63]. The main data sources reported for direct cost estimates were large surveys and insurance claims databases. Self-reported measures for healthcare resource utilisation are known to have limitations with validity of the data due to recall bias [64, 65]. Since many episodes of back pain are recurrent and short lived [66], resource utilisation may crucially be under-reported in survey-based studies. In studies using insurance claims data, the claims might be subject to co-pays, and deductibles or the insurance coverage may vary from plan to plan or from employer to employer. Moreover, since cost estimations are conducted in relation to insured individuals, generalisation of the findings to the wider population may not be appropriate.

The number and type of cost components reported in the studies were highly heterogeneous for both the direct and indirect cost categories. Some studies focussed only on major cost components of consultations, prescriptions, and hospitalisations, whereas others also

considered services such as diagnostic imaging, physiotherapy, and accident and emergency. This meant that significant discrepancies existed in reported costs, and studies with partial estimates of healthcare costs could not be compared with those reporting full COI estimates.

The way in which healthcare resource utilisation is valued could also impact on the reported costs. Most studies used a direct method of summing up back pain related costs which underestimated the true cost of back pain compared to an incremental cost approach. The incremental approach was more comprehensive and accounted for costs that would otherwise have been missed such as costs due to comorbidities resulting from back pain. Valuation of indirect costs using the human capital and friction cost approaches also resulted in widely different estimates with the HC method consistently producing significantly higher costs than the FC method. However, unlike the direct cost estimation methods, the justification for the choice of one method over the other is not clear and there is ongoing debate as to the best method [67–69]. A review and assessment of the evidence suggests that a pragmatic approach is to use both the HC and FC approaches as sensitivity analyses [70].

## Strengths, limitations, and comparison with other studies

This is the first review aimed at assessing the methodologies used in COI studies of back pain. The previous review [1], which focussed on synthesising results rather than methodologies, was from 2008 and the findings were slightly limited by only including studies published in the previous 10 years. A major strength of this review was that large number of studies were included that varied greatly by country and methodology, and the search was conducted with no restriction on publication date. Methodological differences that may not always be apparent but resulting in significant influence on COI estimates of back pain were revealed by this review. A potential limitation was that this review only considered studies conducted in developed countries and published in English. Nevertheless, the findings of this review can be considered robust given that such a large number of studies were examined.

It was noteworthy that several studies did not explicitly describe their methodologies posing challenges to assess them. This lack of clarity is also confirmed by other systematic reviews [1, 71–73], and appears to be a common feature of COI studies. There was also a lack of consensus and guideline on the use of methodologies which may make the analyses prone to underestimation or overestimation of the true costs of the illness. This finding is consistent with that of other systematic reviews of COI analyses, and was not restricted to back pain [74–77]. Another limitation was that the studies did not explicitly report on costs of complications such as revision surgery for infection which are a major source of treatment costs post-surgery. No such distinction was made between costs due to complications and other costs. Furthermore, sensitivity analysis does not appear to be a standard practice in current COI studies with only a minority of studies conforming to the norm. Sensitivity analysis is also rarely done in COI studies of other conditions [72, 74].

## Key implications and recommendations

The lack of standardised and validated instrument and research methodology in COI studies meant that researchers must be careful with their terminology, data source, and methodology used for estimation. Certain types of approaches might be more appropriate than others which has implications for replicating and validating a specific study. The methodological considerations highlighted in this review offer practical guidance to researchers, decision makers, and funders in designing future COI studies. The trade-offs in the various methodological options available for performing the calculations and their effects on the resulting cost estimates should

not be underestimated. Stakeholders should consider what degree of accuracy is required in COI analysis to ascertain the appropriate methods that will meet decision makers' needs.

Based on the findings of this review, some important recommendations for good practice can be drawn which may help produce more reliable estimates for the costs of back pain with implications for COI studies in general:

- As back pain imposes an enormous economic burden, costs must be estimated more accurately and inclusively, and a bottom-up approach using an incremental cost method is recommended as base case analysis.

- Identification of cases should be based on a broad consensus on case definition of back pain, and the use of diagnostic codes is preferred.

- Resource utilisation may be better estimated by following up a large sample of patients from electronic health records, and valuation of unit costs should be carefully assessed.

- Patient characteristics, such as comorbidities, should be clearly reported. Where appropriate, separate cost estimates need to be reported for subgroups of patients.

- Researcher should test the sensitivity of the analysis, give detailed descriptions, and discuss limitations of the methodological choices.

- Finally, development of guideline and standardisation of the methodologies used for COI studies may not only enhance the reliability and interpretation of the estimates, but it also enables comparability of the results across studies.

## Conclusions

COI studies may provide important information and serve as a basis for further economic evaluations and allocation of resources. In the absence of widely accepted standards and consensus on methodology, conducting a COI study capable of identifying and measuring the true cost of an illness remains a challenge. Methodological variations and the discrepancies that arise within them have direct impact on the comparability and credibility of COI studies. This review has reported a widespread heterogeneity in the methodologies used, and the substantial direct or indirect cost estimates produced for back pain. By informing the relative importance of this health problem, the information obtained here has important implications on the allocation of scarce resources and other health policy decision making.

This review also highlighted some factors that might have substantial impact on the reported cost estimates. Recommendations about good practice for COI studies of back pain have been suggested based on the findings of this review. These recommendations may help obtain reliable estimates of the true cost of the illness, improve the quality and reporting of the analysis, and provide validity to COI studies.

## Supporting information

**S1 File. Example of search strategy constructed with breakdown of hits obtained.**
(DOCX)

**S2 File. Comparison of the direct and indirect costs of back pain and the cost components included.**
(DOCX)

**S3 File. PRISMA checklist.**
(DOCX)

## Acknowledgments

The authors thank Dr Majid Artus PhD for his expertise on the clinical area, and the comments and feedback received on the draft manuscripts.

## Author Contributions

**Conceptualization:** Gwenllian Wynne-Jones, Tracy Roberts.

**Supervision:** Jesse Kigozi, Gwenllian Wynne-Jones, Alessandra Guariglia, Tracy Roberts.

**Writing – original draft:** Dawit T. Zemedikun.

**Writing – review & editing:** Dawit T. Zemedikun, Jesse Kigozi, Gwenllian Wynne-Jones, Alessandra Guariglia, Tracy Roberts.

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
