## [Decision Letter · Decision Letter 0]

13 Jan 2021

PONE-D-20-18476

Methodological Considerations in the Assessment of Direct and Indirect Costs of Back Pain: A Systematic Scoping Review

PLOS ONE

Dear Dr. Zemedikun,

Thank you for submitting your manuscript to PLOS ONE. After careful consideration, we feel that it has merit but does not fully meet PLOS ONE’s publication criteria as it currently stands. Therefore, we invite you to submit a revised version of the manuscript that addresses the points raised during the review process.

We look forward to receiving your revised manuscript.

Kind regards,

Sandra C. Buttigieg, MD PhD FFPH

Academic Editor

PLOS ONE

Additional Editor Comments:

I have read with great interest your scoping review and found it to be relevant and meaningful. I do however agree with both reviewers that some revisions are needed prior to being accepted for publication. I do invite you to respond to all the comments raised. In particular, I agree with reviewer 2 to make the necessary changes suggested that would satisfy a broad readership that does not necessarily include only health economists.

2. We note that your literature search was performed on November  2019;to allow an up-to-date view of the topic, we would request that the search is updated.

Reviewers' comments:

Reviewer's Responses to Questions

**Comments to the Author**

1. Is the manuscript technically sound, and do the data support the conclusions?

Reviewer #1: Yes

Reviewer #2: Yes

2. Has the statistical analysis been performed appropriately and rigorously? 

Reviewer #1: Yes

Reviewer #2: I Don't Know

3. Have the authors made all data underlying the findings in their manuscript fully available?

Reviewer #1: Yes

Reviewer #2: Yes

4. Is the manuscript presented in an intelligible fashion and written in standard English?

Reviewer #1: Yes

Reviewer #2: Yes

5. Review Comments to the Author

Reviewer #1: The paper covers an important subject and is well written.

I do not see any major issues that would necessitate a major revision.

However, I would ask the authors to address some minor issues:

- The authors repeatedly mention that the unique feature of their study was the analysis of multiple databases (as opposed to the previous analysis that used only one database). What was the benefit of all these additional databases (other than Medline / PubMed)? Please add a sentence either mentioning how many of ultimately analyzed studies would have been missed on simple Medline analysis, or that addition of all other databases did not add anything of value.

- The explanation of the groups A-F/1-8 is included in supplemental material – I suggest moving it into the actual paper as another table; otherwise this is quite unclear.

- I could not find any figure legends – I am sure they are somewhere but not in the reviewed PDF file (unless the one-liners in the text are the actual legends).

- Was main variability noted between the countries or within the individual countries?

- Were there any features that should be absolutely avoided in order to generate valid data?

Reviewer #2: This paper aims to (1) characterise the literature on the costs of back pain to (2) assess the methods used in COI studies and to (3) present national aggregate costs.

I have the following comments:

General

(1) The paper seems to focus on aim (2), but given the primary audience is not health economists, it may be of greater relevance to the readers to give more weight to summarising the findings of the literature, then discussing the methods.

Search Strategy

(1) Why was the search limited to OECD countries? Were there many papers beyond the OECD nations?

Results

(1) Figure 2:

- suggest swapping inpatient and outpatient so the largest is first

- suggest using a table with $ as the percentage representation masks the fact that the indirect costs are so much larger than the health cost

OR

It may be better to simply include a table of the per patient costs reported in each paper including noting what costs were included.

(2) Table 1 seems too far away from the jargon heavy section on page 16 on indirect costs – suggest moving it closer with in the paper

(3) Indirect cost estimation (last line) : Did any studies report confidence intervals or standard deviations?

Discussion

(1) The very high proportion of indirect costs represented by presenteeism (44 - 85%) is worthy of discussion

(2) What are the key implications of the paper that would be relevant to most readers who might refer to the information in their own papers and implications for policy

(3) Explaining why sensitivity analysis is important is probably more important than providing detail on studies of prostate cancer and Alzheimers – it would be sufficient to say that “sensitivity analysis is also rarely done in COI studies of other conditions [69,71]”.

6. PLOS authors have the option to publish the peer review history of their article (what does this mean?). If published, this will include your full peer review and any attached files.

Reviewer #1: No

Reviewer #2: No

---

## [Author Response · Author response to Decision Letter 0]

24 Mar 2021

Response: We have adjusted the manuscript accordingly to meet PLOS ONE’s style requirements.

2) We note that your literature search was performed on November 2019; to allow an up-to-date view of the topic, we would request that the search is updated.

Response: We have now updated the search to February 2021. One additional study found that met the criteria (Alonso-Garcia 2020) has been included in the review and the text, tables and figures have all been updated accordingly.

Reviewer #1: The paper covers an important subject and is well written.

I do not see any major issues that would necessitate a major revision.

However, I would ask the authors to address some minor issues:

- The authors repeatedly mention that the unique feature of their study was the analysis of multiple databases (as opposed to the previous analysis that used only one database). What was the benefit of all these additional databases (other than Medline / PubMed)? Please add a sentence either mentioning how many of ultimately analyzed studies would have been missed on simple Medline analysis, or that addition of all other databases did not add anything of value.

Response: We thank the reviewer for this comment. Given the very large number of citations (8,009) imported from the six databases, it is difficult to quantify exactly how many studies would have been missed (if any) if the additional databases were not used. Although we believe that in being inclusive to multiple databases, the chances of missing vital papers were minimised; we have now revised the relevant sentences in the discussion (strengths and limitations section) to ensure that the benefit was not overstated.

- The explanation of the groups A-F/1-8 is included in supplemental material – I suggest moving it into the actual paper as another table; otherwise this is quite unclear.

Response: We thank the reviewer for this. We agree with the reviewer’s suggestion, and we have now moved the table into the manuscript as Table 1 (in the methods section).

- I could not find any figure legends – I am sure they are somewhere but not in the reviewed PDF file (unless the one-liners in the text are the actual legends).

Response: We apologise about this. The legend was submitted as a separate document and it appears that this did not make it into the pdf document. We have now incorporated the legends for Fig 1 and 2 in the manuscript itself.

- Was main variability noted between the countries or within the individual countries?

Response: We thank the reviewer for the query. The main methodological variabilities were discussed under the section ‘Key methodological challenges’. These appear to have influenced the broad range of cost estimates reported by the studies in different countries or between studies within the same country. Apart from these, we have not noted any major variabilities worthy of reporting in the review.

- Were there any features that should be absolutely avoided in order to generate valid data?

Response: Thank you for another important question. Each method has its own strengths and weaknesses in relations to the data and purpose of the study. As far as we are aware there no particular features that should be absolutely avoided to generate valid data and so we have made that clear in the ‘Summary of findings’ section by adding the following statements: “The validity of each method would be related to the available data and the proposed use of the findings. Hence, this review did not find any particular features that should be absolutely avoided to generate valid data.”

Reviewer #2: This paper aims to (1) characterise the literature on the costs of back pain to (2) assess the methods used in COI studies and to (3) present national aggregate costs.

I have the following comments:

General

(1) The paper seems to focus on aim (2), but given the primary audience is not health economists, it may be of greater relevance to the readers to give more weight to summarising the findings of the literature, then discussing the methods.

Response: We thank the reviewer for the valuable comment. Following the suggestion, we have now amended the manuscript accordingly. We first changed the ‘Reported cost estimates’ subtitle to ‘Summary of findings’ and we moved this section to the beginning of the discussion to give more weight to summarising the findings of the literature before the discussion of the methodological challenges. The ‘summary of findings’ section has also been expanded and strengthened. 

Search Strategy

(1) Why was the search limited to OECD countries? Were there many papers beyond the OECD nations?

Response: An initial scoping search showed that the vast majority of COI studies on back pain were conducted in OECD countries. We found very little beyond these countries, and hence we decided to focus the search on OECD countries where access to healthcare, and the structure of healthcare systems are more comparable amongst these higher income countries. We have now made this clear by adding the following sentence in the ‘Literature search’ section: “We focussed the search on OECD countries where access and structure of the healthcare systems are more comparable amongst these high-income countries”.

Results

(1) Figure 2:

- suggest swapping inpatient and outpatient so the largest is first

Response: We thank the reviewer for this comment. We have now updated the plot so that outpatient cost key is shown first followed by inpatient, pharmaceutical, and other costs.

- suggest using a table with $ as the percentage representation masks the fact that the indirect costs are so much larger than the health cost

OR

It may be better to simply include a table of the per patient costs reported in each paper including noting what costs were included.

Response: We thank the reviewer for this valuable comment. We have subsequently explored the options and have come up with the following improvements. There are already 4 tables in the manuscript now after moving one of the supplementary tables (categorisation criteria) into the manuscript following the reviewer’s advice. Hence we thought this information may be best reported in a supplementary table both due to the number of tables already included and also because most of the content of this table were also shown in Table 3 so this will avoid repetition. We have therefore added a supplementary table (S2 File) which shows the per capita costs and the cost components included as suggested by the reviewer, and we referred to this table in the statement comparing direct and indirect costs (page 20 of the tracked manuscript). We have kept figure 3 as we thought it gave a good representation of the indirect costs and comparison between the cost components, while actual comparison of direct and indirect costs is now given in the supplementary table in addition to what is in Table 3. We hope that this is an acceptable solution.

(2) Table 1 seems too far away from the jargon heavy section on page 16 on indirect costs – suggest moving it closer with in the paper

Response: We thank the reviewer for this comment. We have moved the table into page 14 (labelled Table 3 now), just before giving results of the main methodological characteristics of the included studies.

(3) Indirect cost estimation (last line): Did any studies report confidence intervals or standard deviations?

Response: We thank the reviewer for this important query. We have now clarified this in the results section on page 20 of the tracked manuscript as follows: “Measures of precision or dispersion such as confidence intervals or standard deviations around cost estimates were rarely reported in the included studies, and those that reported were largely limited to studies that applied econometric methods for cost estimation [17, 23, 24, 43]. In addition, the measures given were generally for the sample estimates (average resource use or cost) rather than for the extrapolated national cost estimates.

Discussion

(1) The very high proportion of indirect costs represented by presenteeism (44 - 85%) is worthy of discussion

Response: We thank the reviewer for this comment. We have now incorporated a discussion of this as follows: We moved the sentence ‘This indicates that presenteeism is often underexplored but represents a significant proportion of the indirect costs of illness’ which was a discussion point, from the results section to the discussion section under ‘Reported cost estimates’ and expanded on it as below:

‘However, in the case of indirect costs, our review indicated that presenteeism is often underexplored but represents a significant proportion of the indirect costs of an illness. The significance of presenteeism for the value of lost production were also highlighted in previous literature [54, 69, 70]. A sound methodological framework for the assessment of presenteeism poses a challenge, but the potential impact of presenteeism on costs needs to be included in order to improve the reliability of results [69, 70].’

(2) What are the key implications of the paper that would be relevant to most readers who might refer to the information in their own papers and implications for policy

Response: We thank the reviewer for highlighting this. We have now changed the ‘Recommendations’ subtitle to ‘Key implications and recommendations’ and have added the following paragraph:

“The lack of standardised and validated instrument and research methodology in COI studies meant that researchers must be careful with their terminology, data source, and methodology used for estimation. Certain types of approaches might be more appropriate than others which has implications for replicating and validating a specific study. The methodological considerations highlighted in this review offer practical guidance to researchers, decision makers, and funders in designing future COI studies. The trade-offs in the various methodological options available for performing the calculations and their effects on the resulting cost estimates should not be underestimated. Stakeholders should consider what degree of accuracy is required in COI analysis to ascertain the appropriate methods that will meet decision makers’ needs.”

(3) Explaining why sensitivity analysis is important is probably more important than providing detail on studies of prostate cancer and Alzheimers – it would be sufficient to say that “sensitivity analysis is also rarely done in COI studies of other conditions [69,71]”.

Response: We thank the reviewer for this comment. Following the advice, we have now amended this accordingly. We have removed the previous statements and replace it with ‘Sensitivity analysis is also rarely done in COI studies of other conditions [70, 72]’.

---

## [Decision Letter · Decision Letter 1]

27 Apr 2021

Methodological Considerations in the Assessment of Direct and Indirect Costs of Back Pain: A Systematic Scoping Review

PONE-D-20-18476R1

Dear Dr. Zemedikun,

We’re pleased to inform you that your manuscript has been judged scientifically suitable for publication and will be formally accepted for publication once it meets all outstanding technical requirements.

Kind regards,

Sandra C. Buttigieg, MD PhD FFPH

Academic Editor

PLOS ONE

Additional Editor Comments (optional):

Reviewers' comments:

Reviewer's Responses to Questions

**Comments to the Author**

1. If the authors have adequately addressed your comments raised in a previous round of review and you feel that this manuscript is now acceptable for publication, you may indicate that here to bypass the “Comments to the Author” section, enter your conflict of interest statement in the “Confidential to Editor” section, and submit your "Accept" recommendation.

Reviewer #1: All comments have been addressed

Reviewer #2: All comments have been addressed

2. Is the manuscript technically sound, and do the data support the conclusions?

Reviewer #1: Yes

Reviewer #2: Yes

3. Has the statistical analysis been performed appropriately and rigorously? 

Reviewer #1: Yes

Reviewer #2: (No Response)

4. Have the authors made all data underlying the findings in their manuscript fully available?

Reviewer #1: Yes

Reviewer #2: Yes

5. Is the manuscript presented in an intelligible fashion and written in standard English?

Reviewer #1: Yes

Reviewer #2: Yes

6. Review Comments to the Author

Reviewer #1: All concerns have been addressed - thank you.

The material will be helpful to the future research projects.

Reviewer #2: All comments have been sufficiently addressed by the authors, the paper is an interesting one and will make a good contribution

7. PLOS authors have the option to publish the peer review history of their article (what does this mean?). If published, this will include your full peer review and any attached files.

Reviewer #1: **Yes: **Konstantin V. Slavin, MD

Reviewer #2: No

---

## [Editor Report · Acceptance letter]

29 Apr 2021

PONE-D-20-18476R1 

Methodological considerations in the assessment of direct and indirect costs of back pain: A systematic scoping review 

Dear Dr. Zemedikun:

I'm pleased to inform you that your manuscript has been deemed suitable for publication in PLOS ONE. Congratulations! Your manuscript is now with our production department. 

Kind regards, 

on behalf of

Professor Sandra C. Buttigieg 

Academic Editor

PLOS ONE